# Structural insights into mechanism and specificity of the plant protein O-fucosyltransferase SPINDLY

Li Zhu[1,2], Xiting Wei[1,2], Jianming Cong[1], Jing Zou[1], Lihao Wan[1] & Shutong Xu [1] ✉

*Arabidopsis* glycosyltransferase family 41 (GT41) protein SPINDLY (SPY) plays pleiotropic roles in plant development. Despite the amino acid sequence is similar to human O-GlcNAc transferase, *Arabidopsis* SPY has been identified as a novel nucleocytoplasmic protein O-fucosyltransferase. SPY-like proteins extensively exist in diverse organisms, indicating that O-fucosylation by SPY is a common way to regulate intracellular protein functions. However, the details of how SPY recognizes and glycosylates substrates are unknown. Here, we present a crystal structure of *Arabidopsis* SPY/GDP complex at 2.85 Å resolution. SPY adopts a head-to-tail dimer. Strikingly, the conformation of a 'catalytic SPY'/GDP/'substrate SPY' complex formed by two symmetry-related SPY dimers is captured in the crystal lattice. The structure together with mutagenesis and enzymatic data demonstrate SPY can fucosylate itself and SPY's self-fucosylation region negatively regulates its enzyme activity, reveal SPY's substrate recognition and enzyme mechanism, and provide insights into the glycan donor substrate selection in GT41 proteins.

In organisms ranging from bacteria to protozoans and metazoans, O-linked glycosylation on proteins are crucial mechanisms used by cells to diversify their protein functions and dynamically regulate their signal integration and physiological states[1–4]. Secreted or cell surface proteins are commonly attached with long oligosaccharide chains via O-linkage while a single monosaccharide being O-linked to Ser/Thr residues without further attachment of additional sugar residues is usually observed for nuclear and cytosolic proteins. Aberrant O-linked glycosylations has been linked to diabetes, neurodegenerative diseases and cancers in human[5–9], and numerous developmental disorders in plants[1,4]. O-GlcNAc transferase (OGT), a member of the glycosyltransferase family 41 (GT41), uniquely catalyzes the transfer of *N*-acetylglucosamine from UDP-*N*-acetylglucosamine (UDP-GlcNAc) to target proteins via O-linkage[10,11]. For long time, only ER-localized protein O-fucosyltransferases (POFUTs) in the GT65 family are reported, which are responsible for catalyzing O-fucosylation of secreted or cell surface proteins[12,13]. Recently, another GT41 protein, SPINDLY (SPY), is identified to be a novel nucleocytoplasmic-localized POFUT, which was found modifying transcription factor and chloroplast-localized co-

chaperonin in *Arabidopsis* and nuclear pore proteins in human pathogen *Toxoplasma gondii*[14–17].

The *Arabidopsis* SPY was initially identified as a negative regulator of phytohormone gibberellin (GA) signal transduction as spy mutants display GA overdosed phenotypes including longer hypocotyls, increased stem growth, early flowering and male sterility[18–20]. Later studies show that SPY plays important roles in multiple developmental processes, including the circadian clock[21], phytohormone cytokinin signaling[22,23], plant architecture[24,25], root development[26], abiotic and biotic stresses[27–29]. Because of the similarity in amino acid sequence compared with human OGT, *Arabidopsis* SPY was long considered as an OGT. However, its OGT activity was never biochemically demonstrated. Strikingly, through sensitive MS analysis and in vitro enzyme assays, *Arabidopsis* SPY was then found to be a novel POFUT[17]. DELLA proteins, consisting of GA INSENSITIVE (GAI), REPRESSOR OF *ga1-3* (RGA) and RGA-LIKE1 (RGL1), RGL2 and RGL3, are central integrators in the GA signaling. *Arabidopsis* SPY can high-selectively catalyze O-fucosylation of DELLA proteins and subsequently activate DELLA by promoting its interaction with target proteins[17]. PSEUDO-RESPONSE

[1]Hubei Hongshan Laboratory, College of Life Science and Technology, Huazhong Agricultural University, Wuhan 430070, China. [2]These authors contributed equally: Li Zhu, Xiting Wei. ✉e-mail: xushutong@mail.hzau.edu.cn

REGULATOR 5 (PRR5) is a core circadian clock component. *Arabidopsis* SPY can O-fucosylate PRR5 and facilitate the proteolysis of PRR5, resulting in modulating the circadian clock[16]. *Arabidopsis* SPY is also found responsible for the fucosylation of a chloroplast-localized co-chaperonin CPN20, which negatively regulating ABA signaling in seed germination and seedling development[14].

In animals, dynamic cycling of O-GlcNAc is established by the OGT and O-GlcNAc hydrolase OGA. In contrast, the O-GlcNAc modification in plants are catalyzed by the OGT SECRET AGENT (SEC), however, a specific O-GlcNAc hydrolase has not yet been identified. Balance between O-GlcNAc and O-fucose modifications on plant proteins was proposed as a strategy to overcome the lack of O-GlcNAc hydrolase on the basis of that the O-GlcNAc or O-fucose modifications on specific proteins like DELLA could display opposite effects on their functions[30]. Meanwhile, *spy sec* mutants are embryonic lethal while spy mutants display severe developmental disorders and *sec* mutants possess very subtle phenotypes[20,31], suggesting a certain overlap of function for at least a subset of crucial target proteins. The complex interplay between O-fucose and O-GlcNAc modifications in mediating signaling transduction in plants is still largely unknown.

In fact, SPY-like proteins extensively exist in diverse branches of life, including prokaryotes, protists, algae and all plants, indicating that O-fucosylation by SPY is a common way to regulate intracellular protein functions[1]. Lacking a structure of full-length SPY and information of how it interacts with target proteins have been a major impediment to ultimately illuminate how this novel and functionally important POFUT engages in protein complexes and selects its substrates. In this work, we determined the three-dimensional (3D) structure of *Arabidopsis* SPY/GDP complex using X-ray crystallography. SPY adopts a head-to-tail dimer. In the crystal lattice, the N-terminal loop of one protomer in the symmetry-related SPY dimer B traverses the active site of one protomer in SPY dimer A, forming a 'catalytic SPY'/GDP/'substrate SPY' complex. *Arabidopsis* SPY's self-fucosylation region and its functional importance are examined in vitro. Structure-based analysis also reveals SPY's molecular mechanism and provides insights into the glycan donor substrate selection in GT41 proteins.

## Results

### Crystallization and biochemical characterization of SPY

The full-length *Arabidopsis* SPY was recombinantly expressed in *Escherichia coli* cells and purified via multi-step processes (see details in methods). Although it shows good homogeneity, we failed to produce any crystals after a systematic effort. Surface exposed free cysteine was previously reported to suppress protein crystallization in some cases[32]. We thus generated a series of SPY mutants with cysteine (C) mutated to serine (S). Crystals of engineered SPY^C645S appears in about one month after the crystallization drop was set up. The crystals belong to $P2_12_12_1$ space group and the structure was finally refined to a 2.85 Å resolution (Table 1).

Sequence alignment of SPY proteins from different species reveals that C645 in *Arabidopsis* SPY is not conserved (Supplementary Fig. 1), implying that mutation of C645 will not affect SPY's structure or function. Size exclusion chromatography with inline multi-angle light scattering (SEC-Mals) analysis showed that both wild type SPY and SPY^C645S adopt dimeric form in solution (Supplementary Fig. 2). To assess the in vitro enzyme activity, wild type SPY or SPY^C645S was incubated with GDP-fucose and then subjected to HPLC analysis respectively. Similar amount of GDP was detected in both reaction groups, indicating the C645S mutation have little impact on the enzyme activity of SPY (Supplementary Fig. 3a). Additionally, we prepared an *Arabidopsis* DELLA truncation covering the SPY-targeting region (designated as DELLA^1–205). Taking advantage of Aleuria aurantia lectin (AAL) that specifically bind to fucose moieties on glycoproteins (see details in methods), fucosylation of DELLA^1–205 catalyzed by wild type SPY and SPY^C645S was detected by western blotting with AAL

(Supplementary Fig. 3b). Taken together, sequence alignment, SEC-Mals analysis and enzymatic assays collectively demonstrate that the crystal structure of engineered SPY^C645S represents the conformation of wild type SPY.

### The structure of SPY/GDP complex

In agreement with our SEC-Mals analysis result (Supplementary Fig. 2), two SPY molecules form a homodimer in the crystal structure (Fig. 1). The two protomers in the SPY dimer adopt almost identical conformation with a root mean squared deviation (RMSD) of 0.94 Å in the corresponding Cα positions (Fig. 1a, b). The N-terminal region (residues 1–43) forms a long flanking loop while only F20-S37 in one protomer and G23-S37 in another protomer were observed in the crystal structure. Eleven tetratricopeptide repeats (TPRs) (residues 44–430) make up a superhelical conformation. Two extra helices and the following loop (residues 431–482) connect the TPR region to the catalytic domain. The catalytic domain (residues 483–847) comprises the N-terminal lobe (N-Cat) and the C-terminal lobe (C-Cat). Both the N-Cat and the C-Cat assume Rossmann-like folds typical of GT41 subfamily members. A short C-terminal region following the catalytic domain (residues 848–914) is invisible in the structure due to less of density map. GDP is observed on the surface of C-Cat adjacent to the interface with N-Cat (Fig. 1b and Supplementary Fig. 4).

The overall structure of *Arabidopsis* SPY monomer shows little similarity with that of human ER-localized protein O-fucosyltransferase POFUT1/2[33,34] but most resembles that of human OGT[35,36] (Supplementary Fig. 5). The catalytic regions in *Arabidopsis* SPY and human OGT can be superimposed well with a RMSD of 1.81 Å for 380 aligned Cα atoms (Supplementary Fig. 5a). *Arabidopsis* SPY and human OGT

**Table 1 | Statistics of X-ray diffraction and structure refinement**

|  | SPY/GDP complex (PDB code 7Y4I) |
|---|---|
| **Data collection** | |
| Space group | $P2_12_12_1$ |
| Cell dimensions | |
| $a, b, c$ (Å) | 127.7, 130.7, 141.4 |
| α, β, γ (°) | 90, 90, 90 |
| Resolution (Å) | 50–2.85 (2.95–2.85)[a] |
| $R_{merge}$ (%) | 20.8 (303.1)[a] |
| I / σI | 9.4 (0.7)[a] |
| $CC_{1/2}$ | 0.994 (0.527)[a] |
| Completeness (%) | 100 (100)[a] |
| Redundancy | 13.4 (13.6)[a] |
| **Refinement** | |
| Resolution (Å) | 35.3–2.85 (2.90–2.85)[a] |
| No. reflections | 55,644 |
| $R_{work}$ / $R_{free}$ | 21.3 (35.0)[a]/27.0 (39.4)[a] |
| No. atoms | |
| Protein | 12779 |
| Ligand | 28 |
| Water | 5 |
| $B$-factors | |
| Protein | 86.8 |
| Ligand/ion | 171.9 |
| Water | 17.1 |
| R.m.s. deviations | |
| Bond lengths (Å) | 0.003 |
| Bond angles (°) | 0.6 |

[a]Values in parentheses are for highest-resolution shell.

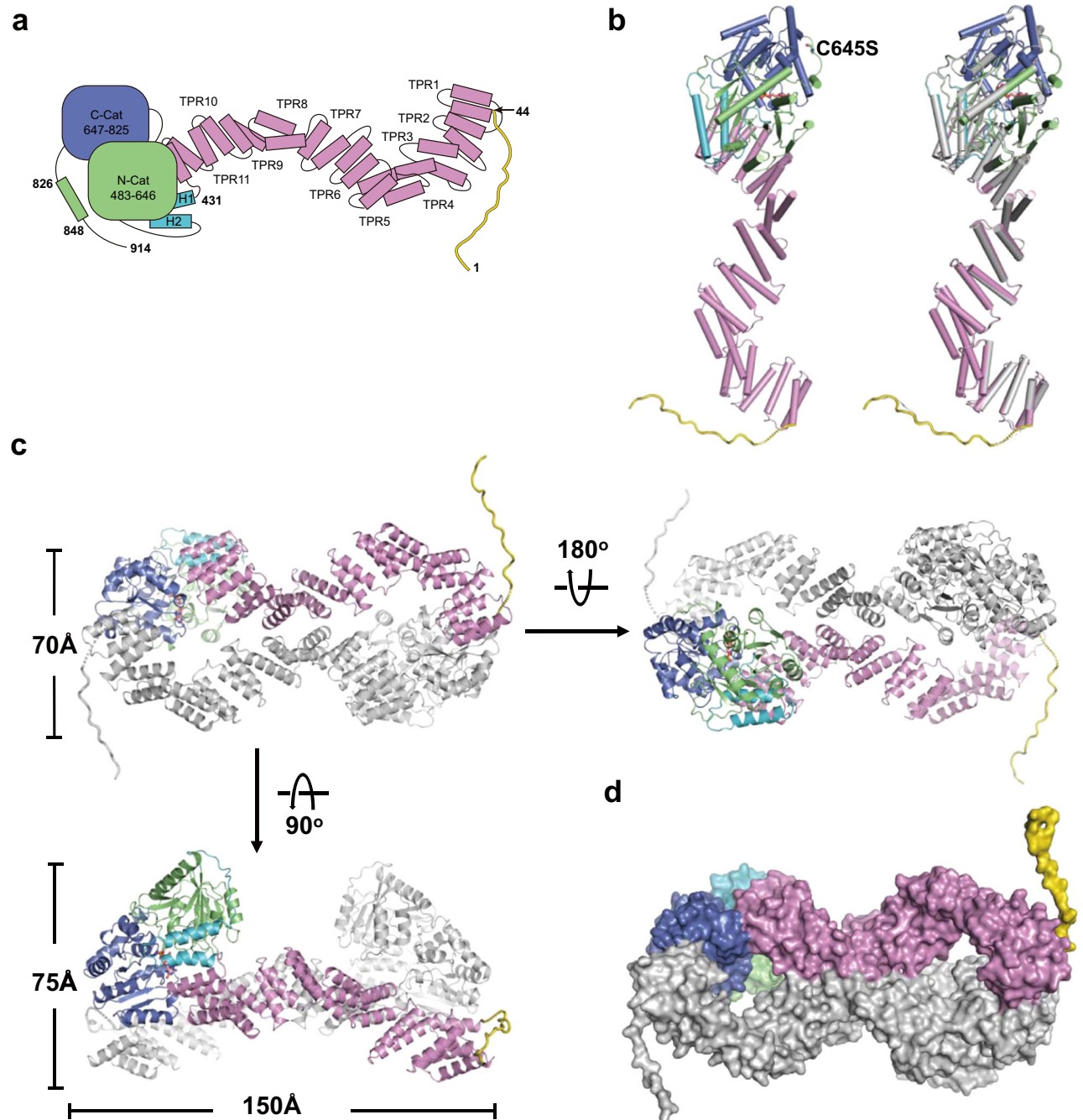

**Fig. 1 | Structure of SPY/GDP complex. a** Schematic diagram of *Arabidopsis* SPY. The N-terminal loop, the TPR domain, the connector region, the N-Cat lobe and the C-Cat lobe are colored in yellow, pink, turquoise, green and purple, respectively. **b** Left panel is the structural overview of the SPY/GDP complex in cartoon representation. The C645S and GDP are shown in ball-and-stick model and colored in magenta and salmon, respectively. Right panel is the structural comparison of the two protomers in the SPY dimer. One protomer is colored the same as the schematic in **a** and the other protomer is colored in gray. **c** Overall structure of SPY dimer. One protomer is colored the same as the schematic in **a** and the other protomer is colored in gray. **d** Surface representation of the SPY dimer.

contain a similar number of TPR units, both forming a two-layer right-handed superhelix. However, the TPR superhelix in *Arabidopsis* SPY is ~120 Å long and ~30 Å wide while that in human OGT is relatively shorter and plumper (Supplementary Fig. 5b). Although superposition of the C-terminal four TPR repeats, TPRs 8-11 in *Arabidopsis* SPY and TPRs 10–13 in human OGT, yields a small RMSD (1.45 Å for 100 aligned Cα atoms), the rest parts of the two TPR domains show large conformational differences (Supplementary Fig. 5b). Detailed investigation of the *Arabidopsis* SPY structure reveals that the TPR4 and TPR7

form unique conformations with their first helices being two turns longer than canonical TPR helices (Supplementary Fig. 5c, d), resulting in a smaller spiral width and larger spiral length of the superhelix. In addition, using the catalytic domains as a reference, the TPR superhelix of *Arabidopsis* SPY rotates about 30° relative to the equivalent region of human OGT (Supplementary Fig. 5a). Furthermore, notable conformational differences exist in the connector regions of the two proteins. Compared with that in human OGT, the connector region in *Arabidopsis* SPY is much shorter and comprises one less α helix

(Supplementary Fig. 5a). Previous studies on human OGT reveals that the connector region is crucial for its catalysis and the TPR domain mainly functions in substrate binding[36–38]. Therefore, these results indicate that *Arabidopsis* SPY has a distinctive catalytic and substrate recognition mechanism.

## SPY exhibits unique dimerization mode

In the crystal structure, two SPY molecules interact with each other in a head-to-tail manner, forming a symmetrical dimer with a length of ~150 Å, a width of ~70 Å and a height of ~75 Å (Fig. 1c). The dimer gives rise to two distinct interfaces: interface 1 (and interface 1'), which is mainly generated by TPR1 from one protomer and C-Cat lobe from another protomer (Supplementary Fig. 6a, red square); and interface 2, which is generated by TPRs 7-9 from the two protomers antiparallelly contacting each other (Supplementary Fig. 6a, blue square). The dimer interface in total is ~1809 Å$^2$, burying ~5% of each monomer surface area (Fig. 1d). The dimerization of SPY is maintained by hydrogen bonds (Y289-R342', N252-N344', K58-S713', K58-V716') and hydrophobic contacts (Y49, W292, V316, Q319, L320, F324, and F654). Details of the interactions are shown in Supplementary Fig. 6b.

A 3D structural homology search with the program DALI[39] revealed that SPY's head-to-tail dimeric conformation has not been previously observed within other GT41 protein structures. Human OGT assumes a domain-swapped dimer mediated only by two TPR units with relatively small interface (~560 Å$^2$) in both crystal structure[36] and cryo EM structure[35] (Supplementary Fig. 7). The TPR domains in human OGT dimer is considered of inherent flexibility based on that the N-terminal 11.5 TPR units of two human OGT protomers form different conformation in the crystal structure[36] while the TPRs 1–5 exhibited high heterogeneity and were very hard to be modeled in the cryo EM structure[35]. The flexibility of the TPR domain is believed to allow human OGT adopting different overall conformations upon interaction with different binding partners[36]. On the contrary, it is highly unlikely to happen in the case of *Arabidopsis* SPY. Although the two TPR regions in *Arabidopsis* SPY dimer form two looser and longer superhelices as compared to that in human OGT dimer, they are locked in a fixed and almost identical conformation thanks to dimer interactions (Fig. 1c).

## The structure of 'catalytic SPY'/GDP/'substrate SPY' complex and self-fucosylation of SPY

HPLC-based enzymatic assay showed that free GDP was detected after SPY incubated with GDP-fucose (Supplementary Fig. 3a), indicating that either SPY can hydrolyze the GDP-fucose or SPY is a substrate for itself. During the model building of our *Arabidopsis* SPY/GDP complex structure, strikingly, we found that the N-terminal loop of one protomer in a symmetry-related SPY dimer B, encompassing the residues from F20* to S37*, wrapped around the concave surface of the superhelix and traverse the active site of one protomer in the SPY dimer A (Fig. 2a and Supplementary Figs. 8 and 9). More interestingly, S21* in dimer B is located close to the phosphate groups of the GDP in one active site of dimer A, strongly suggesting it is a target residue for fucosylation. Altogether, SPY dimers A and B seem to form a 'catalytic SPY'/GDP/'substrate SPY' complex.

To assess whether SPY can be self-fucosylated, we performed enzymatic assay by western blotting with biotinylated AAL. In the presence of GDP-fucose, fucosylation of SPY was detected (Fig. 2b). Carefully examination of the SPY sequence reveals the N-terminal loop is rich of Ser/Thr residues (Supplementary Fig. 1). We then generated two SPY truncations: SPY$^{\Delta1-14}$ and SPY$^{\Delta1-42}$ with the N-terminal 14 and 42 residues deleted, respectively. As shown in Fig. 2b, SPY$^{\Delta1-14}$ possess similar fucosylation level as the full-length SPY, however, SPY$^{\Delta1-42}$ totally abolished the self-fucosylation. These results demonstrate that the fucosylation modification occur in the region between V15 and Q42. We next individually mutated S21 and another Ser/Thr residue

closest to it, S24, to alanine. In presence of GDP-fucose, the fucosylation of SPY$^{S21A}$ and SPY$^{S24A}$ was significantly reduced compared with wild type SPY, suggesting that the two residues are the major sites for self-fucosylation (Fig. 2c).

We next investigate whether self-fucosylation of SPY acts as an important factor governing its function. It is common that multiple substrates compete for the active site resulting in a competitive inhibition of the enzyme, which raises the possibility that the N-terminal loop of SPY could compete with other substrates and function as a negative regulator of its enzyme activity. Remarkably, SPY$^{\Delta1-42}$ showed increase glycosylation activity toward DELLA$^{1-205}$ compared with the full-length SPY and SPY$^{\Delta1-14}$, whereas mutation of S21 and S24 did not affect SPY's fucosylation activity (Fig. 2d, e). These results demonstrate that the self-fucosylation region (V15-Q42) but not O-fucosylation on SPY can negatively regulate its activity.

## Protein substrate recognition in SPY

The 'catalytic SPY'/GDP/'substrate SPY' complex assumes a 'T wrench' shape with the N-terminal extended loop in one substrate SPY molecule (hereafter the substrate loop) being tightly coupled to the TPR superhelix in one catalytic SPY molecule (hereafter SPY) and penetrating directly into the active site (Figs. 2a and 3a). In previous studies, overexpression of SPY's TPR domain created a dominant-negative effect that conferred a spy-like phenotype[40,41]. Yeast-two-hybrid and co-immunoprecipitation assays showed that deletion of the TPR domain in SPY totally abolished its interaction with PRR5, which is one of the core circadian clock components[16]. These results suggest that SPY's TPR domain is required for substrate binding, which is consistent with our structural observations.

Close examination of the structure of the 'catalytic SPY'/GDP/'substrate SPY' complex reveals that the substrate loop stretches across more than 50 Å and is most embraced by the C-terminal 6 TPRs (TPRs 6–11) of SPY through extensive hydrogen bonds. The TPR superhelix in SPY consists of two layers, the inner layer is formed by the first helices in each TPR repeat while the outer layer is formed by the second ones (Supplementary Fig. 5c). A continuous asparagine ladder throughout the inner layer of the TPR superhelix, including N300 in TPR8, N333 and N334 in TPR9, N367 and N368 in TPR10, N402 in TPR11, together with N436 in the connector region recognize the backbone of the substrate loop by forming a hydrogen-bond network (Fig. 3a and Supplementary Fig. 5d). Besides, the side chains of Y299 in TPR8 and R408 in TPR11 interact with the backbone of the substrate loop through hydrogen bonds. To verify the residue interactions observed in our structure, we generated two alanine-substitution mutants, including SPY$^{N5}$ (N300A, N334A, N367A, N368A, N402A) and SPY$^{N7}$ (SPY$^{N5}$ combined with R408A, N436A). In vitro enzymatic assay showed that the self-fucosylation of SPY$^{N5}$ and SPY$^{N7}$ were totally abolished (Fig. 3b). Furthermore, in presence of GDP-fucose, the fucosylation of DELLA$^{1-205}$ by SPY$^{N5}$ or SPY$^{N7}$ is undetectable (Fig. 3c). Altogether, our structural observations and enzymatic data demonstrated that the asparagine ladder in the TPR domain is critical for substrate binding.

Beyond the TPR domain and the connector region, S496 in the N-Cat also interacts with the substrate loop (Fig. 3a). However, S496 is more likely to stabilize the orientation of the reactive serine (S21*) in the substrate loop, which will be discussed later. Surprisingly, the side chains of the substrate loop rarely participated in the interaction with SPY. S37* is the only one in the substrate loop, whose side chain is hydrogen-bonded to K231 in TPR6 and D266 in TPR7 in SPY, suggesting K231 and D266 play a role in substrate selectivity. To assess the importance of K231 and D266, we evaluated the activity of SPY$^{K231A}$, SPY$^{D266A}$, SPY$^{K231A/D266A}$ (Fig. 3b, c). Unexpectedly, mutations of K231 and D266 did not affect the self-fucosylation but had a significant impact on the fucosylation of DELLA$^{1-205}$. SPY$^{K231A}$ and SPY$^{K231A/D266A}$ showed substantially decreased activity while SPY$^{D266A}$ showed increased activity

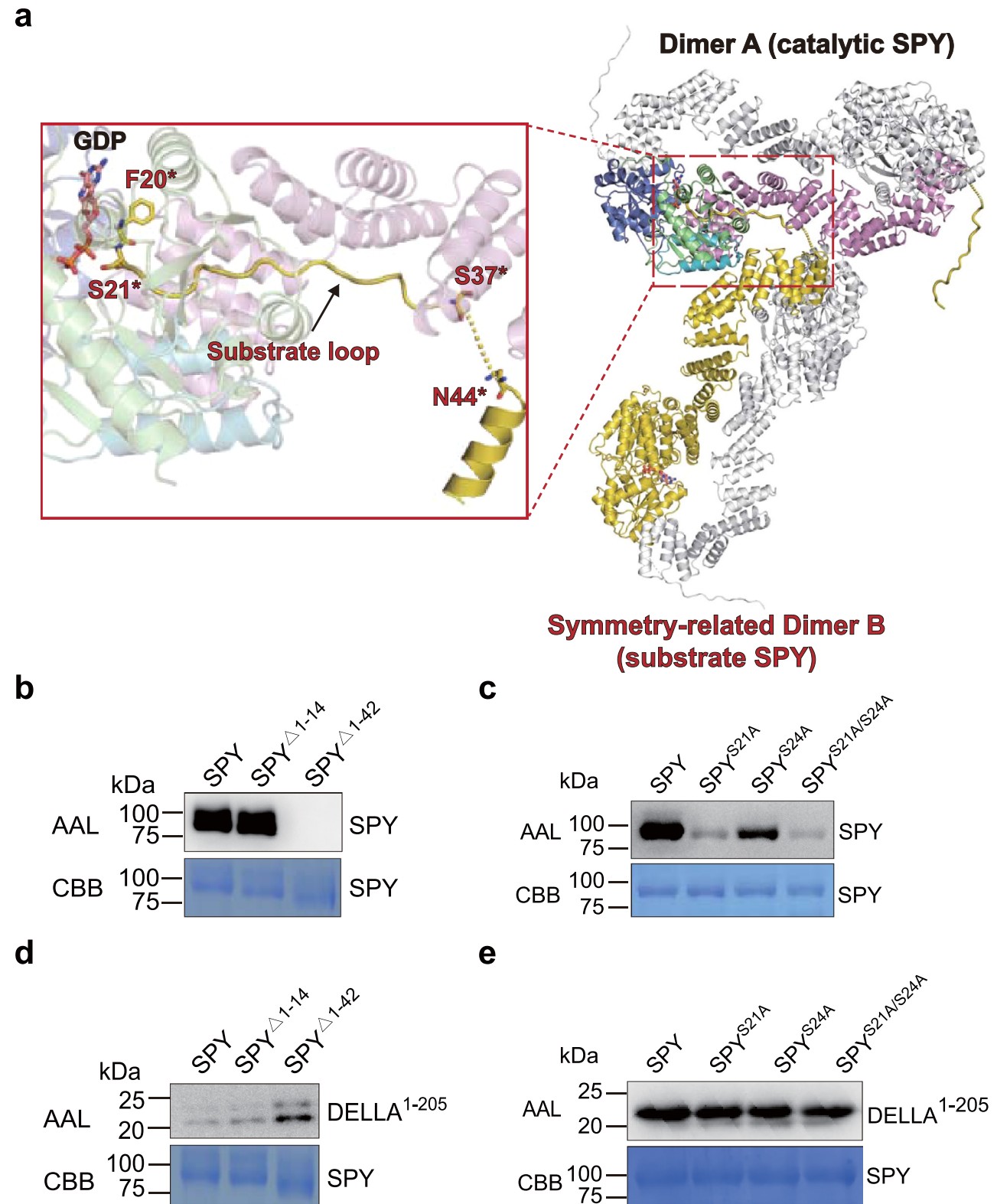

toward DELLA$^{1-205}$. Furthermore, compared with SPY$^{K231A}$, the double mutant showed increased activity. Taken together, our data demonstrated that K231 and D266 drive SPY's protein substrate selection.

**Active site and insights for the catalytic mechanism of SPY**

In the 'catalytic SPY'/GDP/'substrate SPY' structure, the active site locates in a pocket sandwiched by the two catalytic lobes, with its opening partially being covered by the C-terminus of the TPR domain

and the connector region (Fig. 4a). The GDP moiety is mainly coordinated by the C-Cat (Fig. 4b). The side chains of N661, K665 and T748 make hydrogen bonds with the phosphates of the GDP. The side chain of H728 makes a π−π stacking interaction with the purine ring of GDP. Besides, the binding of GDP also involves the backbones of I722, L723, T747 and T749 in SPY. Although Y744 does not form direct interaction with GDP, its side chain is close to a region that potentially accommodate the fucose moiety, hence Y744 probably stabilize the fucose

**Fig. 2 | SPY can fucosylate itself. a** Overview of an 'catalytic SPY'/GDP/'substrate SPY' complex in the crystal lattice. In SPY dimer A (catalytic SPY), the color coding is the same as in Fig. 1c. In symmetry-related SPY dimer B (substrate SPY), the two protomers are colored in yellow and gray, respectively. A close-up view of the 'catalytic SPY' and 'substrate SPY' interface is shown in red square. In 'substrate SPY' molecule, residues before F20* and in the between of S37* and N44* are invisible in the structure. S21* is located close to the GDP. **b** Self-fucosylation of full-length SPY and N-termini truncations. Upper panel shows western blotting with biotinylated AAL. Lower panel shows Coomassie blue staining. Experiments were independently repeated three times with similar results. Uncropped images of western blotting membranes and gels are available as source data. **c** Self-fucosylation of SPY single-point mutants. Upper panel shows western blotting with biotinylated AAL. Lower

panel shows Coomassie blue staining. Experiments were independently repeated three times with similar results. Uncropped images of Western blotting membranes and gels are available as source data. **d** Fucosylation of DELLA by full-length SPY and N-termini truncations. Upper panel shows western blotting with biotinylated AAL. Lower panel shows Coomassie blue staining. Experiments were independently repeated three times with similar results. Uncropped images of western blotting membranes and gels are available as source data. **e** Fucosylation of DELLA by SPY single-point mutants. Upper panel shows western blotting with biotinylated AAL. Lower panel shows Coomassie blue staining. Experiments were independently repeated three times with similar results. Uncropped images of western blotting membranes and gels are available as source data.

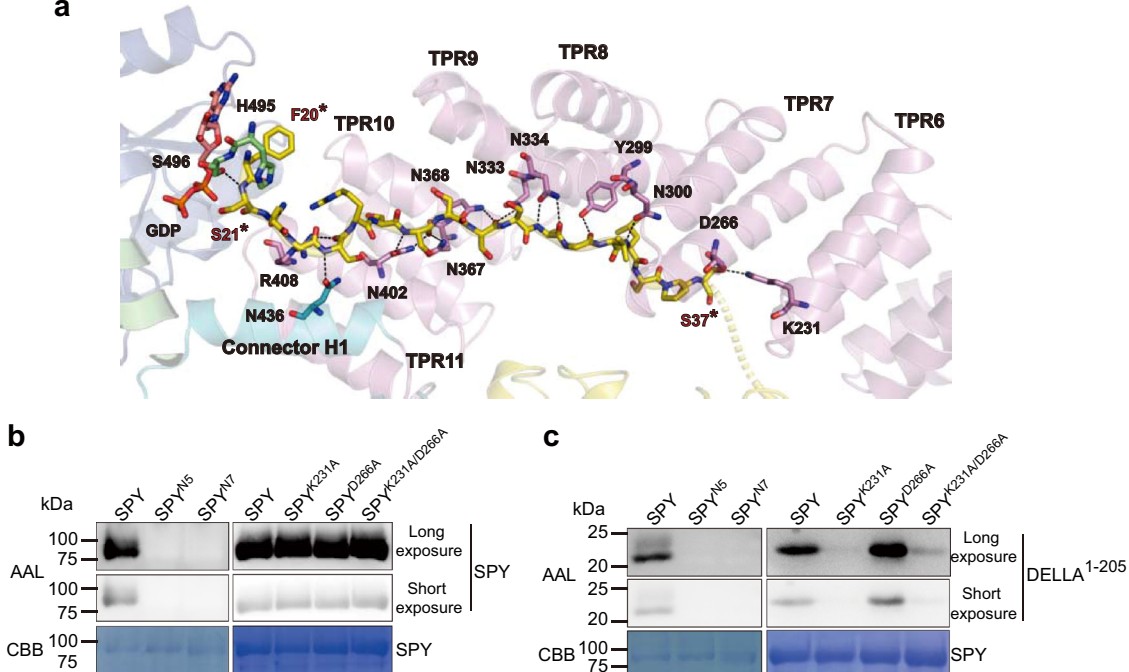

**Fig. 3 | Substrate binding in SPY. a** Close-up view of the substrate binding in the 'catalytic SPY'/GDP/'substrate SPY' complex. The 'catalytic SPY' molecule is shown as cartoon. Residues of interest and GDP in the 'catalytic SPY' are shown as sticks and labeled in black. The substrate loop in the 'substrate SPY' molecule is shown as sticks. F20* and S21* in the substrate loop are labeled in red. Black dotted lines indicate hydrogen bonds. The color coding is the same as in Fig. 2a. **b** Self-fucosylation of SPY variants. SPY^N5 contains N300A, N334A, N367A, N368A and N402A. SPY^N7 is SPY^N5 combined with R408A and N436A. Upper panel (long exposure) and middle panel (short exposure) show western blotting with

biotinylated AAL. Lower panel shows Coomassie blue staining. Experiments were independently repeated three times with similar results. Uncropped images of Western blotting membranes and gels are available as source data. **c** Fucosylation of DELLA by SPY variants. Upper panel (long exposure) and middle panel (short exposure) show western blotting with biotinylated AAL. Lower panel shows Coomassie blue staining. Experiments were independently repeated three times with similar results. Uncropped images of Western blotting membranes and gels are available as source data.

moiety. To assess the functional roles of these residues, we individually substituted each of them with alanine and assayed the enzyme activity of the mutants (Fig. 4c, d). Mutations of N661 and H728 slightly impaired the self-fucosylation of SPY and hardly affect the enzyme activity towards DELLA$^{1-205}$, indicating a dispensable role of them. In contrast, the SPY^S496A, SPY^Y744A and SPY^T748A mutants possessed a significantly decreased enzyme activity, demonstrating a key role of the three residues in the fucose transfer reaction. Mutation of K665 completely lost the activity, indicating that K665 are essential for the catalytic reaction.

As demonstrated by our structural and biochemical data, S21* in the substrate loop is a major target residue to be fucosylated by SPY. The side chain of S496 form hydrogen bonds with the backbone amine of S21*. Mutations of S496 dramatically weakened the enzymatic activity (Fig. 4c, d). Altogether, the structural and biochemical data demonstrate that S496 stabilizes the orientation of S21* and probably helps orient the substrate loop. S21* together with the preceding

residue F20* in the substrate loop lie over the GDP moiety, subsequently covering the nucleotide-sugar binding site and blocking access to it. A similar feature was observed in the ternary complex of the closest structural homolog, human OGT (Fig. 4b and Supplementary Fig. 5e). Reinforced by kinetic studies, the ordered bi-bi catalytic mechanism has been demonstrated for human OGT[38]. Presumably SPY would also follow an ordered bi-bi catalytic mechanism in which GDP-fucose binds before the polypeptide substrate.

General base catalysis is generally thought to have a role in the glycosyltransferase mechanism, in which an aspartate, glutamate or histidine typically functions as the catalytical base[42,43]. Such kinds of residues rarely exist in the active site of SPY, with H495 being closest to S21* hydroxyl. The imidazole side chain of H495 stacks with the D491 carboxylate, indicating that H495 and D491 may function as a catalytic dyad (Fig. 4b). However, although SPY^H495A and SPY^H495F are both catalytically dead, SPY^D491A retains activity (Fig. 4c, d). Considering mutation of H495 to alanine would free up more space in the active site

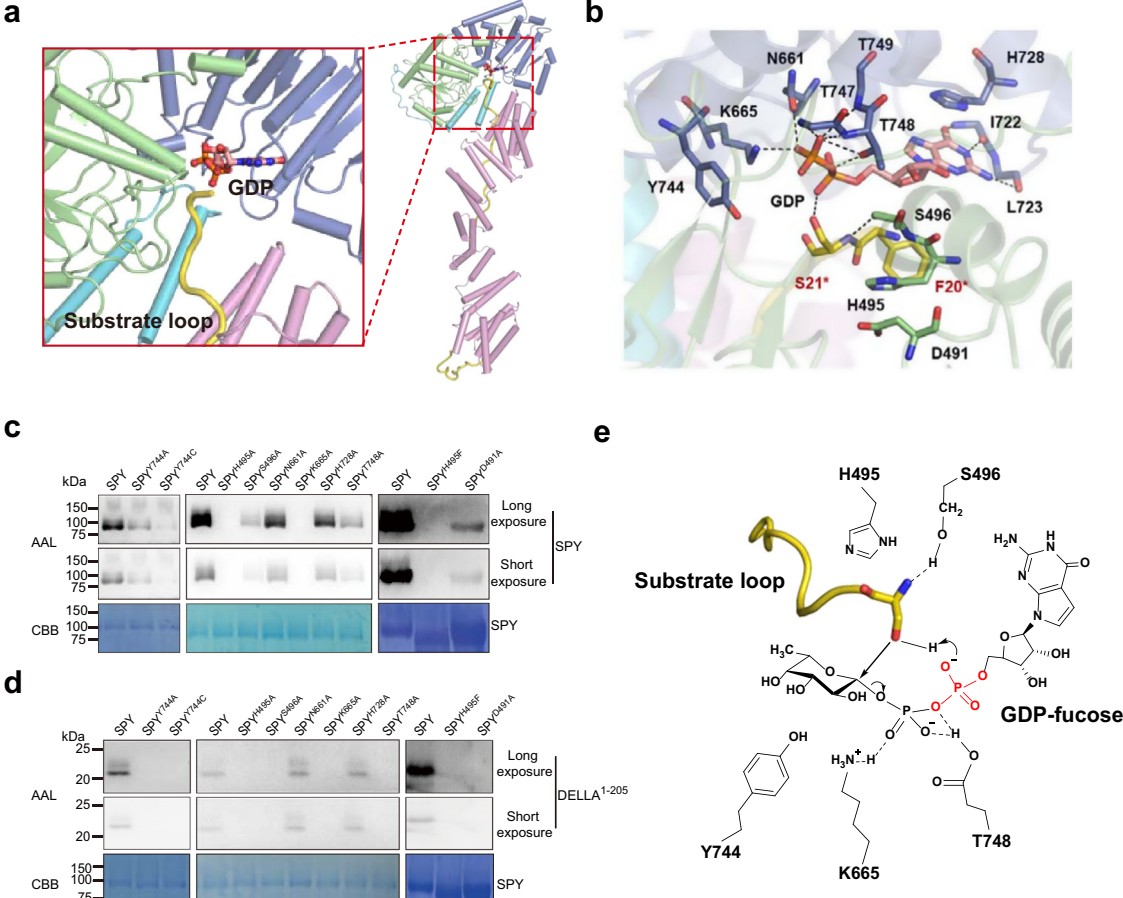

**Fig. 4 | Active site and catalytical mechanism of SPY. a** Cartoon rendering of the active site in the 'catalytic SPY'/GDP/'substrate SPY' complex. GDP is shown as stick. The color coding is the same as in Fig. 2a. **b** Close-up view of the active site in the 'catalytic SPY'/GDP/'substrate SPY' complex. Residues of interest are shown as sticks. For simplicity, the side chains of I722, L723, T747 and T749 are omitted as they stabilize the GDP with their backbones. Residues in the 'catalytic SPY' and the substrate loop are labeled in black and red, respectively. Black dotted lines indicate hydrogen bonds. **c** Self-fucosylation of wild type SPY and mutants carrying point mutations in the active site. Upper panel (long exposure) and middle panel (short exposure) show western blotting with biotinylated AAL. Lower panel shows Coomassie blue staining. Experiments were independently repeated three times with similar results. Uncropped images of Western blotting membranes and gels are available as source data. **d** Fucosylation of DELLA by wild type SPY and mutants carrying point mutations in the active site. Upper panel (long exposure) and middle panel (short exposure) show western blotting with biotinylated AAL. Lower panel shows Coomassie blue staining. Experiments were independently repeated three times with similar results. Uncropped images of Western blotting membranes and gels are available as source data. **e** Proposed catalytic mechanism of SPY. The peptide is depicted in yellow loop with the reactive serine shown as stick. The α-phosphate in the GDP-fucose could act as the catalytic base. H495 probably contributes to maintain the structure of the active site. S496 stabilizes the orientation of the reactive serine and the substrate loop. K665 and T748 stabilize the GDP moiety. Y744 potentially stabilize the fucose moiety.

while phenylalanine is more hydrophobic and slightly bigger than histidine, the effects of the H495 mutations are possibly due to structural reasons. Additionally, the S21* hydroxyl rotamer is not suitable for engaging in a hydrogen bond with either the D491 carboxylate or the H495 imidazole ring (Fig. 4b). Further inspection of the 'catalytic SPY'/GDP/'substrate SPY' complex reveals that none of the residues in SPY is within 4 Å of S21* hydroxyl. Altogether, SPY may not harbor the catalytic base. Notably, the acceptor substrate hydroxyl donates a hydrogen bond to one of the α-phosphate oxygen in the donor substrate while the α-phosphate lacks any interactions with positively charged side chains, hence this phosphate could serve as the catalytic base as proposed earlier for human OGT[44] (Fig. 4e).

### Implications on the glycan donor substrate selection of GT41 proteins

Although the overall structure of the catalytic domain in *Arabidopsis* SPY is quite similar to that in human OGT, strikingly, superposing the structures of the *Arabidopsis* 'catalytic SPY'/GDP/ 'substrate SPY' complex and the human OGT/UDP-GlcNAc/HCF-1[1-26] complex (PDB 4N3C)[45] using the catalytic domains as a reference, the side chain of Y744 in *Arabidopsis* SPY (equivalent to C917 in human OGT) and the side chain of reactive Ser21* on the substrate loop exhibit severe steric hindrance with the GlcNAc moiety of UDP-GlcNAc bound in human OGT (Fig. 5a), indicating that the active site in *Arabidopsis* SPY is not commodious enough to simultaneously bind UDP-GlcNAc and the substrate peptide. Compared with the GlcNAc, the fucose is a relatively smaller hexose with a hydroxyl instead of *N*-acetyl group attached at C2 and a methyl instead of hydroxymethyl group attached at C5 of the pyranose ring (Supplementary Fig. 10), hence it can be appropriately accommodated in the active site of SPY. As mentioned earlier, the SPY[Y744A] mutant possess a significantly decreased fucose-transferring activity (Fig. 4c, d). We further mutated Y744 to cysteine and evaluated the enzyme activity. Both the self-fucosylation and the fucosylation of DELLA[1-205] by SPY[Y744C] can hardly be detected (Fig. 4c, d), again showing that Y744 is critical for the POFUT activity of *Arabidopsis*

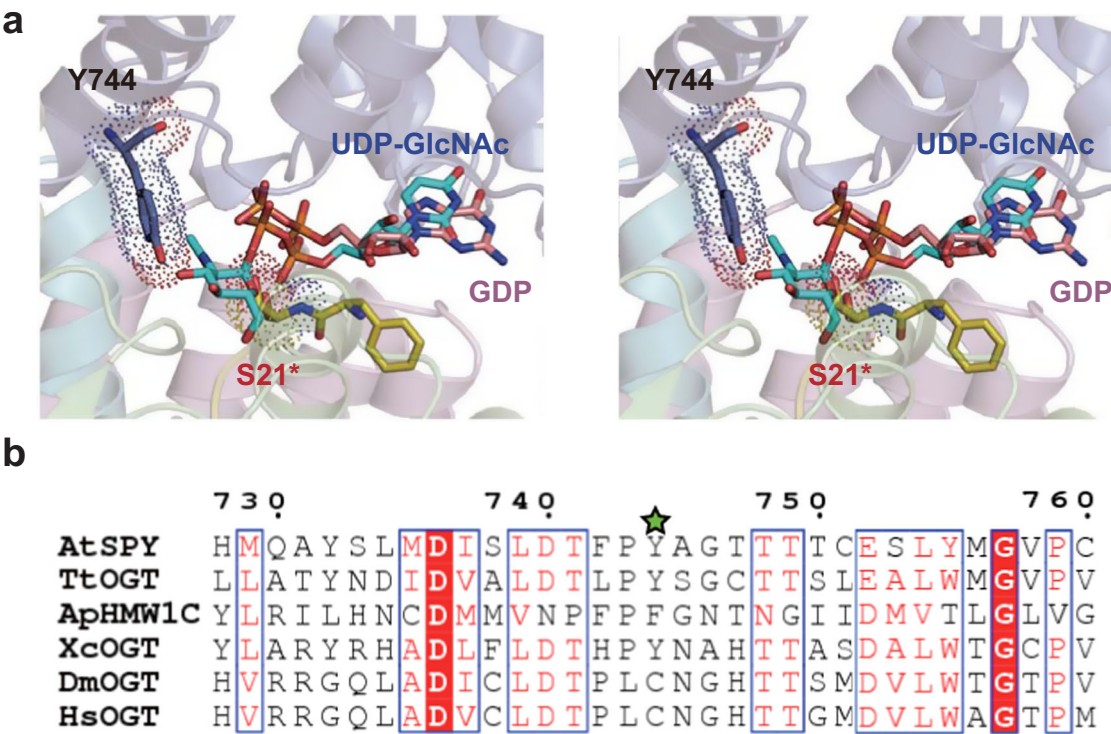

**Fig. 5 | Comparison of AtSPY and other GT41 proteins. a** Structural superposition of the active sites in the *Arabidopsis* 'catalytic SPY'/GDP/'substrate SPY' complex and the human OGT/UDP-GlcNAc/HCF-1¹⁻²⁶ complex (PDB code 4N3C) was performed using the catalytic domains as a reference. For *Arabidopsis* 'catalytic SPY'/GDP/'substrate SPY' complex, GDP and some residues discussed in the text are shown as sticks. Specially, Y744 and S21* are also shown in dot representation. For simplicity, for the human OGT/UDP-GlcNAc/HCF-1¹⁻²⁶ complex only the UDP-GlcNAc is shown as stick and colored in cyan. The figure is in wall-eyed stereo. **b** Structure-based sequence alignment of *Arabidopsis* SPY (AtSPY), *Thermobaculum terrenum* OGT (TtOGT; PDB 5DJS), *Xanthomonas campestris* OGT (XcOGT; PDB 2JLB), *Actinobacillus pleuropneumoniae* HMW1C protein (ApHMW1C; PDB 3Q3H), *Drososphila melanogaster* OGT (DmOGT; PDB 5A01) and human OGT (HsOGT; PDB 4N3C). Y744 in AtSPY is indicated using a green star. Only a short fragment containing Y744 in AtSPY is shown here. Sequence alignment of the connector regions and catalytic domains is shown in Supplementary Fig. 11.

SPY. Collectively, our results rationalize how SPY is highly selective to GDP-fucose.

So far, besides human OGT, only structures of four other members in GT41 family have been determined, including human OGT orthologues in *Drosophila* (DmOGT)[46], the plant pathogen *Xanthomonas campestris* (XcOGT)[47] and the soil thermophile *Thermobaculum terrenum* (TtOGT)[48], and *Actinobacillus pleuropneumoniae* HMW1C protein (ApHMW1C)[49]. We then conducted structure-based sequence alignment of the catalytic domains in these proteins together with *Arabidopsis* SPY and human OGT (Fig. 5b and Supplementary Fig. 11). The active site in DmOGT is identical to that in human OGT and it indeed possesses O-GlcNAc activity in vitro[46]. In XcOGT, the active site is more similar to that in human OGT rather than *Arabidopsis* SPY, however, the residue equivalent to C917 in human OGT (Y744 in *Arabidopsis* SPY) is Y463. Interestingly, different from human OGT modifying thousands of proteins[10,11], no O-GlcNAc activity was detected for XcOGT against bacterial cell lysates or short peptide substrates. Instead, XcOGT O-GlcNAcylated a single substrate in *Arabidopsis* cell lysates[47]. It is highly possible that UDP-GlcNAc is not a very appropriate glycan donor substrate for XcOGT. In TtOGT and ApHMW1C, the residue equivalent to Y744 in *Arabidopsis* SPY is Y419 and F517, respectively. Notably, TtOGT did not exhibit any O-GlcNAc activity in vitro, and MS analysis of the *T. terrenum* proteome failed to identify any O-GlcNAcylated proteins[48]. ApHMW1C used UDP-α-D-glucose as glycan donor to create N-glycosidic linkages with glucose and galactose at asparagine residues and O-glycosidic linkages between glucose residues on HMW1 protein[49]. Glucose is apparently smaller than GlcNAc (Supplementary Fig. 10). Taken together, in GT41 family, a single residue located equivalent to Y744 in *Arabidopsis* SPY

determines the size of the active site and may drive the glycan donor substrate selection.

## Discussion

Protein O-fucosylation by SPY plays a key role in regulating plant development[1]. It may be a common way to regulate intracellular protein functions as SPY-like proteins extensively exist in diverse organisms. Here we have determined the crystal structure of *Arabidopsis* SPY/GDP complex. In the crystal lattice, we unexpectedly captured a 'catalytic SPY'/GDP/'substrate SPY' complex conformation, in which GDP and the N-terminal loop of a symmetry-related 'catalytic SPY' molecule are bound in the active site of the 'substrate SPY' molecule. On the basis of our structural and biochemical data, we found that *Arabidopsis* SPY can fucosylate itself at the N-terminal loop and the major target residues are S21 and S24 (Fig. 2a–c). Previous studies show that many glycosyltransferases such as human OGT, XcOGT, NleB1 from several enteropathogenic bacteria species could glycosylate themselves and the self-glycosylation are crucial for their biological function[47,50,51]. O-glycosylation modification of proteins can usually either determine subcellular localization, enhance or disrupt protein-protein interactions, or induce a conformational change[30]. In this case, the fucosylation modification is attached to a flanking loop, thus it is unlikely to induce a big conformation change of *Arabidopsis* SPY. Intriguingly, we further identified that the N-terminal flexible loop but not the O-fucosylation modification in *Arabidopsis* SPY can act as a negative regulator to its activity (Fig. 2d, e), which might be a regulatory mechanism of SPY-dependent O-fucosylation in *Arabidopsis*. Although the N-terminal regions in SPY orthologs from different organisms share low sequence similarity, some of them contain lots of

Ser/Thr residues (Supplementary Fig. 1), indicating that the self-fucosylation may be a common feature of the SPY proteins. In supportive of this idea, *Toxoplasma gondii* SPY was also reported capable of self-fucosylation[15].

*Arabidopsis* SPY exist as head-to-tail dimers in solution and the crystal structure. Different from the domain-swapped human OGT dimer which harbors one relatively small interface and adopts a flexible conformation, the head-to-tail SPY dimer contains three contacting areas and forms a rigid structure. Either mutation of the self-fucosylation sites or deletion of the N-terminal flexible loop or fucosylation modification do not alter the dimeric form of SPY (Supplementary Fig. 6c). We also assessed the behavior of SPY in presence of protein substrate. Although DELLA1-205 did not co-migrated with SPY in the gel filtration assay, which is probably due that DELLA1-205 has a poor affinity for SPY, only SPY dimer but no other form was detected in the mixture (Supplementary Fig. 6d). Altogether, it is very likely that dimer is the functional form of SPY. Mutation of K665 in the active site renders the enzyme inactive. Compared with the SPY$^{WT}$/SPY$^{WT}$ homodimer, the SPY$^{WT}$/SPY$^{K665A}$ heterodimer shows substantially weaker activity toward itself and DELLA$^{1-205}$ (Supplementary Fig. 6e, f), suggesting that SPY catalyzes protein substrates in 2:2 stoichiometry.

Considerable effort has been dedicated to improved understanding of substrate recognition of GT41 family, yet structural studies of truncated proteins have only yielded incomplete snapshots. Here we have described a structure of full-length SPY in complex with GDP and a 'substrate SPY', which provides a complete structural view of the interactions between the TPRs and the protein substrate (Fig. 3). A continuous asparagine ladder throughout the inner layer of the TPR superhelix and the connector region recognize the backbone of the substrate loop. Notably, most of the key asparagines conservatively occupy position 5 and 6 in the consensus of TPRs 8-11 while position 6 in the consensus of TPR 5 and TPR 6 are similarly occupied with asparagines in *Arabidopsis* SPY (Supplementary Fig. 5d), implying a potential role of TPR 5 and TPR 6 in recognizing other substrates of SPY. Previously an asparagine ladder was also identified to be important for substrate binding in human OGT[52]. Similarly, most of these asparagines in human OGT occupy position 6 in the TPR consensus (Supplementary Fig. 5d). Nevertheless, the asparagine ladder in human OGT stretches across almost the entire TPR region (TPR 3-TPR 14). In *Arabidopsis* SPY, N333 and N367 which also form hydrogen bonds with the backbone of the substrate loop occupy position 5 in the consensus of TPR9 and TPR10, however, the equivalent residues in human OGT are serines. Therefore, the substrate protein binding modes of *Arabidopsis* SPY and human OGT are not quite the same. In *Arabidopsis* SPY, the key asparagines are evenly distributed in several TPRs and conservatively locate in the inner layer of the TPR superhelix, forming a slender and sinuous substrate-contacting surface. In this context, the SPY-interacting sequences would be likely to reside in unstructured regions in target proteins.

Compared with human OGT in which three aspartate residues in the C-terminal two and half a TPR units (TPRs 12–13.5) recognize the substrate peptide side chains and are identified to be important for its protein substrate selection[37,51], *Arabidopsis* SPY acquire the protein substrate selectivity using a totally different mechanism, where two residues in the middle of the TPR superhelix (K231 in TPR6 and D266 in TPR7) play a key role (Fig. 3a). Mutation of K231 to alanine substantially impaired the fucosylation of DELLA$^{1-205}$ while mutation of D266 to alanine on the contrary enhanced that. Previous study revealed that the SPY-targeting region within DELLA$^{1-205}$ contained poly S/T[17] and multiple of these residues could be O-fucosylated by SPY. Presumably mutations of K231 and D266 altered the O-fucosylation sites in DELLA. Sequence alignment of SPY orthologs in diverse living systems showed that the key residues for protein substrate recognition in *Arabidopsis* SPY are highly conserved (Supplementary Fig. 1), indicating that the SPY proteins share the same substrate recognition mechanism.

The 'catalytic SPY'/GDP/'substrate SPY' complex structure and mutagenesis studies suggest a substrate-assisted catalysis, where the catalytic base is not provided by the enzyme but likely by the α-phosphate on the glycan donor substrate. This differs from ER-localized POFUTs and other fucosyltransferases. For POFUT1 and POFUT2, an active site asparagine undergoing tautomerization and a glutamate residue were proposed to act as the catalytic base, respectively[34,53,54]. Human α1,6-fucosyltransferase (FUT8) is the only enzyme responsible for core-fucosylation, by which a fucose is transferred from GDP-fucose to the innermost GlcNAc residue of N-linked glycans. A glutamate residue has also been proposed to act as the catalytic base for FUT8[55]. Actually, the catalytic region in *Arabidopsis* SPY most resembles that in human OGT. H498, H558 and Y841 in human OGT were initially proposed as candidate bases[38,47,56,57], but were later observed to locate too far away from the acceptor hydroxyl[44,58]. D554 in human OGT was put on the stage because it indirectly interacted with the acceptor hydroxyl via a chain of water molecules[58], however, human OGT$^{D544A}$ retained activity[44]. The equivalent residues of H498, D554, H558 and Y841 in human OGT are L439, D491, H495 and A664 in *Arabidopsis* SPY, respectively (Supplementary Fig. 5a). Apparently, leucine and alanine could not enable the deprotonation of the acceptor hydroxyl. D491 and H495 are adjacent in the structure. The coupled D491/H495 is unlikely to act as the general base because their side chains were positioned too far away from the acceptor hydroxyl and mutation of D491 did not abrogate the catalysis (Fig. 4c, d). More recently, the α-phosphate acting as the catalytic base has been revealed for human OGT based on cumulative crystallographic snapshots and biochemical probes[44], which coincidentally reinforces SPY's substrate-assisted catalysis proposed here.

Previously, twenty spy mutants were identified to exhibit pleiotropic phenotypes, some of which are related to elevated GA response (longer hypocotyls, increased stem growth, early flowering and male sterility) while others are not (such as altered circadian rhythms, cytokinin response, and plant architecture)[17,18,20]. Except that the spy-4 is an RNA null allele, others all contain mutations in the region from TPR7 to the end, which is coincidentally responsible for the substrate recognition and catalysis (Figs. 2 and 3 and Supplementary Fig. 12). The spy-1/2/8 contain an in-frame 23 amino acid deletion (M354-Q376) in TPR9 helix 2 and TPR10 helix 1 while the spy-7 contains replacement of I390-A392 with F in TPR10 helix 2. The spy-6/10/11 (G268E) and spy-9 (G268R) contain a single amino acid substitution of G268, which locates on TPR7 helix 1 with the Cα atom being close to the TPR7 helix 2. Mutating G268 to an amino acid with big side chain (E or R) probably will push TPR7 helix 2 away from TPR7 helix 1. Therefore, it is highly possible that spy-1/2/8, spy-7, spy-6/10/11 and spy-9 would cause conformational changes in the TPR region and affect the recognition of protein substrates by SPY. In fact, the POFUT activity of spy-8 was previously reported to be only 7.3% that of the wild type[17]. The spy-3 (G593S), spy-5 (C845Y), spy-12 (G570D), spy-13/14 (T572M), spy-15 (E567K), spy-16/17 (R815W) and spy-19 (K665M) each contain a single amino acid substitution in the catalytic domain. In the ternary structure of SPY, K665 is hydrogen-bonded with the GDP. Our enzymatic data and previous study[17] have demonstrated that mutation of K665 would totally abrogate its activity. E567, G570, T572 and G593 locate in the loops around the active site. Although no direct interaction with GDP or the substrate loop was observed for these residues, it is possible that mutations of these residues will alter the structure of the active site and subsequently affect the enzyme activity. Consistent with this idea, spy-15 (E567K) was found inactive[17]. R815 and C845 are surface exposed and locate on the C-terminal two helices that are far away from the active site. Therefore, R815W and C845Y are unlikely to impair the enzyme activity. Spy-18 contains deletion of L782-S914. Although these residues mainly form three helices and the connecting loops on the surface of the catalytic domain, we cannot exclude the possibility that such a deletion might affect protein stability and

subsequently impair SPY's activity. Residues of E848-S914 are invisible in our 'catalytic SPY'/GDP/'substrate SPY' complex structure, indicating they are of high structural flexibility and are dispensable for the activity. Therefore, the enzyme activity of spy-20 with a fame shift of T880-S914 probably is not affected.

## Methods

### Molecular cloning
The coding frames of full-length SPY (At3g11540) and DELLA (AT2G01570.1) fragment (spanning residues 1-205) were amplified using appropriate primer pairs from the cDNAs of *Arabidopsis thaliana*. Sequences for primers are listed in Supplementary Data 1. Each gene was inserted into the pET28sumo vector that attaches a 6× His tag plus a SUMO tag at the N-terminus or the pGEX-6p-1 vector that attaches a GST tag at the N-terminus. Constructs of the SPY mutants containing point mutations were generated using the Site-Directed Mutagenesis method or Gibson Assembly method. All the constructs were verified by DNA sequencing.

### Protein expression and purification
The plasmids were transformed into *E. coli* BL21 (DE3) cells. The transformed bacterial cells were grown at 37 °C to $OD_{600}$ of 0.6, and then the protein expression was induced with 0.3 mM isopropyl-β-D-thiogalactoside (IPTG) at 16 °C overnight. The cells were collected, resuspended in a lysis buffer (20 mM Tris, pH 7.9, 500 mM NaCl, 10% Glycerol, 1 mM PMSF, and 5 mM Benzamidine), and lysed by a high-pressure cell disrupter. Target protein was collected from the supernatant and firstly purified over $Ni^{2+}$ affinity resin. After that, the N-terminal 6× His and SUMO tags attached in SPY proteins were cleaved by Ulp1 protease and removed using a second step $Ni^{2+}$ affinity purification. Target protein without tag was further purified by gel filtration on a Superdex 200 column (GE Healthcare). The purified proteins were of high purity (above 95%) as analyzed by SDS-PAGE, and were concentrated to about 10 mg/ml in a storage buffer (20 mM Tris pH 7.9, 300 mM NaCl, 0.5 mM TCEP) for structural and biochemical studies.

Specifically, to investigate the SPY: protein substrate stoichiometry in the catalytic reaction, we prepared recombinant SPY$^{WT}$/SPY$^{WT}$ homodimer and SPY$^{WT}$/SPY$^{K665A}$ heterodimer using the following strategy. 6His-SUMO-tagged SPY$^{WT}$ or 6His-SUMO-tagged SPY$^{K665A}$ were co-expressed with GST-tagged SPY$^{WT}$ in *E. coli* cells. The SPY$^{WT}$/SPY$^{WT}$ homodimer and SPY$^{WT}$/SPY$^{K665A}$ heterodimer were collected by a GST affinity purification, follow by a $Ni^{2+}$ affinity purification. GST tag and 6His-SUMO tag were removed by 3C and Ulp1 proteases, respectively.

### Gel filtration assay
The Superdex 200 Increase 10/300 GL column (GE Healthcare) were used for the SEC analysis. The assay was performed with an injection volume of 0.5 mL, and the buffer containing 50 mM Tris pH 7.9, 300 mM NaCl, 0.5 mM TCEP, 2 mM Benzamidine. All the samples were incubated at 4 °C for 6 h. The peak fractions were analyzed by SDS-PAGE followed Coomassie Brilliant Blue staining. Particularly, the self-fucosylated SPY sample was also analyzed by Western blotting with biotinylated AAL.

### SEC-Mals assay
Wild-type SPY and engineered SPY$^{C645S}$ (~500 µg) were independently loaded onto a Superdex 200 increase 10/300 column (GE Healthcare) connected to a HELEOS multi-angle light scattering instrument (WYATT Technology). All assays were performed in the buffer containing 20 mM Tris pH 7.9, 300 mM NaCl, 0.5 mM TCEP. The figure was drawn using Origin 8.0.

### Crystallization
Crystallization experiments were performed using the sitting drop vapor diffusion methods at 16 °C by mixing the protein with an equal volume of reservoir solution. For obtaining crystals of SPY in complex with GDP or GDP-fucose, the protein was incubated with GDP or GDP-fucose at a molar ratio of 1:10 for 1 h at 4 °C prior to crystallization. We were unable to obtain any crystals of SPY/GDP-fucose complex. In contrary, crystals of SPY in complex with GDP were grown under the condition of 0.1 M bicine pH 8.6, 24% Polyethylene glycol 1,500, and 10% (v/v) 2-Propanol. Crystals were flash frozen using a cryoprotectant consisting of the crystallization buffer and 5% Glycerol.

### Data collection and structure determination
X-ray diffraction data were collected at beamline BL17U and BL19U of Shanghai Synchrotron Radiation Facility[59,60], and processed using HKL2000[61] with cross-correlation to determine the diffraction limit[62] to a resolution of 2.85 Å. The structure was solved by the molecular replacement method using Phenix[63] and a structure predicted using AlphaFold v2.0 system[64] as the search model. Model building was performed using Coot[65] and structure refinement was carried out using Phenix. Stereochemistry of the structure models was analyzed using Molprobity[66]. Structural analysis was carried out using the PISA server[67]. All graphics were generated using Pymol (http://www.pymol.org). Statistics of the structure refinement and the structure models are summarized in Table 1.

### In vitro SPY activity assay by HPLC
The in vitro enzyme activity of wild type and mutant SPY were analyzed using a sensitive HPLC method described previously[68] with slight modification. Specifically, the reaction mixture consisted of 5 µg tag-removed enzyme, 60 µM GDP-fucose, 50 mM Tris pH 8.0, 300 mM NaCl, and 5 mM $MgCl_2$ in a total volume of 40 µl. The reactions were undertaken at room temperature for 1 h and then quenched at 95 °C for 90 s. HPLC analyses were performed with a InertSustain C18 column (5 µm particle size, 250 mm × 4.6 mm, GL science) on an EClassical 3100 series apparatus, equipped with a high-pressure constant current pump (P3100), a well-plate autosampler (S3100), a thermostatted column compartment (O3100) and a variable wavelength detector (DAD3100). The mobile phase consisted of phosphate buffer (10 mM phosphate buffer, pH 3.95) running at a flow rate of 1 mL/min, and the detector was set at 254 nm. The standard GDP and GDP-fucose were used as references.

### In vitro SPY activity assay by western blotting with biotinylated AAL
The in vitro enzyme activity of wild type and mutant SPY were analyzed using western blotting with biotinylated AAL described previously[69] with slight modification. Briefly, for assessing the self-fucosylation of SPY, the reaction mixture consisted of 5 µg tag-removed enzyme, 60 µM GDP-fucose, 50 mM Tris pH 8.0, 300 mM NaCl, and 5 mM $MgCl_2$ in a total volume of 40 µl; for assessing the fucosylation of DELLA$^{1-205}$ by SPY, the reaction mixture consisted of 5 µg tag-removed enzyme, 20 µg DELLA$^{1-205}$, 60 µM GDP-fucose, 50 mM Tris pH 8.0, 300 mM NaCl, and 5 mM $MgCl_2$ in a total volume of 40 µl. The reactions were undertaken at room temperature for 1 h and then quenched at 95 °C for 90 s. Eight microliters of the reaction buffer was subjected to SDS-PAGE and blotted using biotinylated AAL (Vector Labs, cat# B-1395) with a dilution of 1:1000 and HRP-labeled Streptavidin (Sangon, cat# B110053) with a dilution of 1:5000 as the primary and secondary antibodies, respectively. Blots were developed for detection by ChemiScope5600 (Clinx).

## Data availability
The data that support this study are available from the corresponding author upon request. Atomic coordinates of *Arabidopsis* SPY/GDP complex have been deposited in the Protein Data Bank (PDB) under accession number 7Y4I. Atomic coordinates of previously published structures used in this study are also available in PDB under accession numbers 7NTF, 4N3C, 4AY6, 5DJS, 2JLB, 3Q3H and 5A01. Source data are provided with this paper.

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

## Acknowledgements

We thank staffs at the BL17U and BL19U beamline of the NCPSS at Shanghai Synchrotron Radiation Facility for assistance with data collection. We thank the Center for Protein Research, Huazhong Agricultural University, for facilities support. We thank X. Wang and J. Hu for providing the *Arabidopsis* cDNAs. We thank J. Wang for the critical comments on the manuscript. This work was supported by the National Key Research and Development Program (Grant No. 2022YFA0912100 to S.X.), the National Natural Science Foundation of China (Grant No. 32000900 to S.X.), the Fundamental Research Funds for the Central Universities (Program No. 2662018PY056 to S.X.).

## Author contributions

S.X. conceived of the project. S.X., L.Z. and X.W. designed and interpreted the experiments; L.Z., X.W., and J.Z. performed the experiments; J.C. and L.W. collected the X-ray diffraction data and determined the structure. S.X. wrote the manuscript with support from all authors.

## Competing interests

The authors declare no competing interests.
