## [Peer Review File · Nature Communications]

REVIEWER COMMENTS

Reviewer #1 (Remarks to the Author):

The crystal structure of the Arabidopsis nucleocytoplasmic protein O-fucosyltransferase SPY is reported. The work provides mechanistic insights into SPY function by identifying amino acids involved in protein O-fucosylation. SPY was found to self-modify two serines located near the N-terminus. Deletion of the region containing these residues reduces SPY activity suggesting that self-modification may regulate activity. The results generally support the authors' conclusions. The work provides an important foundation for increasing our our understanding of the glycosyltransferase family 41 function and how these enzymes regulate plant development.

Comments: [L1] [SEP] Line 28 and lines 249-251. The data does not support that O-fucosylation negatively regulates SPY. Since the role of self-fucosylation was tested by deleting the region that is modified (Fig. 2d), the results only show that AA 1-42 negatively regulate SPY. To more directly address the role of O-fucose modification, the activity of the S21A, S24A mutants and an S21A / S24A double mutant should be determined.

Lines 125-127. While the hydrolysis assay supports that C645S SPY is a fully functional enzyme, the evidence would be stronger if C645S SPY was shown to have wild-type activity toward a protein substrate such as DELLA.

Lines 125-127. Consider reinforcing the argument that C645 is a surface residue by highlighting its location on a structure shown in one of the figures. Consider pointing out that that C645S is not conserved (Supp Fig 10), which further supports the argument that mutating it will not affect activity.

Line 284. I cannot see any signal for SPYN5 and N7 in the figure. Is there a signal or is it undetectable?

A number of SPY missense mutations affecting plant hormone and other responses are reported in the literature. It would increase the impact and utility of this work if the authors showed where these mutation map on the structure and discuss if the locations suggest that enzyme activity is affected.

Minor comments:

Line 113. serial should be series

Line 118. Please spell out size exclusion chromatography multi angle light scattering (SEC-MALS).

Line 158. Consider large rather than dramatical.

Line 201 Unlike should be unlikely.

Line 222. Replace can with that.

Line 224. Delete truly.

Line 232. ... modification occur in the region between V15 and Q42.

Line 263. Delete excellently.

Reviewer #2 (Remarks to the Author):

This is a very nice manuscript that describes the first crystal structure of SPY, a plant fucosyltransferase that fucosylates protein substrates. The relevance of this structure is of utmost importance because it mainly describes for the first time how the TPRs interact with potential protein substrates (in this case SPY itself). This has never been visualized at the atomic level, and this structure exemplifies this. Therefore, this work offers a significant novelty to merit publication in Nature Communications.

However, major and minor comments would need to be addressed to improve the quality of the manuscript.

Major comments:

- The structure raises a major question that needs to be addressed. For example, how does the dimer behaves in the presence of protein substrates? This needs to be addressed to understand this enzyme's behavior in solution and to rule out whether a monomeric or other form might be present. Their trapped structure might suggest that it is likely an inactive conformation of the enzyme leading to self-fucosylation. Would the dimer be the functional form to recognize protein substrates? Biophysical experiments could show this by incubating SPY with different protein substrates that need SPY TPRs for optimal interaction. For example, they could use PRR5 as a protein substrate. In addition, if the dimer holds in the presence of PRR5, would this be 2:2 (SPY:protein substrate) stoichiometry or 2:1 stoichiometry?

- The authors should check the oligomerization state of SPY once it fucosylates itself. Will the self-fucosylated SPY still be a dimer? Also, check the oligomerization state of SPY truncations and mutants S21A and S24A. Will these mutants and truncations still be a dimer?

- Define SPYN5 and SPYN7 in the text, not only in the figure legend. The experiments with these mutations (the asparagine ladder mutations) are very nice and exemplify previous biophysical experiments of OGT with similar Asn residues as crucial in recognition of protein substrates. The authors should check if these mutations also kill the activity on other protein substrates such as PRR5 or large protein substrates. Check also the activity of K231 or D266 mutants on protein substrates such as PRR5.

- Are these Asn residues conserved with the Asn residues in the human OGT (see PMID: 33709700)? I guess not but it would be nice to compare the positions of these Asn residues in SPY with the Asn residues in the human OGT.

- The mechanism for SPY is not clear. The authors suggest that H495 is the catalytic base. Is this residue conserved with the human OGT? Note that for the human OGT, it was proposed earlier that a His residue was the catalytic base. Then, two further manuscripts suggested that the alpha phosphate (PMID: 23103942) or a chain of water molecules in which one of them interacts with an Asp residue (PMID: 23103939) could act as potential catalytic bases. This should be discussed in the manuscript and the H495 should be clarified if it occupies the same position or similar positions to His498 or His558 in the human OGT. Therefore, new figures showing a proper comparison with the human OGT active site should be shown in the manuscript. In addition, they should compare all these mechanisms with the ones described for PoFUT1, PoFUT2, FUT8, etc (PMIDs: 26854667, 34868727 and 32080177). SPY as a PoFUT should be compared with very distant PoFUTs such as PoFUT1 and PoFUT2 and other fucosyltransferases (FUT8), which also interact with proteins. Finally, molecular dynamics simulations should be performed with GDP-fucose and SPY-peptide to determine the distance of H495 to the acceptor Ser residue of the peptide during the simulations.

- According to the crystallography table, the structure is of lower resolution since the CC1/2 at high resolution is around 0.2. The cutoff for the highest resolution shell should have a CC1/2 around 0.5. Therefore, the data must be rescaled to render a CC1/2 around 0.5 in the highest resolution shell.

This will clearly decrease the current resolution, but it will reflect better the resolution of the structure.

- Show the maps for the Fo-Fc in Supplementary Figure 3. This map is more relevant to see the quality of the density.
- Discussion is too short and should be enlarged, taking into account the catalytic mechanism and also exploiting that this structure is the first one showing interactions between the TPRs and a peptide (here, the N-terminus of SPY).

Minor comments:

- Self-glycosylation is not novel since this occurs in many glycosyltransferases such as human OGT, NleB1, etc. See, e.g., PMID: 32411621. Yet, their finding is very interesting. Cite some of these papers.
- For the ordered bi-bi catalytic mechanism, the authors need to perform additional kinetic experiments and/or NMR experiments to demonstrate this mechanism.
- sentence in lines 37 and 38 does not make much sense. "Different from secretion proteins". Rewrite this sentence.
- What do they mean with aberrant O-linked monosaccharide glycosylation and its linking to diabetes and other diseases? Aberrant or truncated O-glycans are linked to O-glycosylation in the Golgi or E.R. pathway and not to O-glycosylation in the cytosol or in the nucleus. Clarify this.
- Line 65. Define DELLAs. Define also GA signal transduction (line 55).
- Supplementary Figure 11. N-GlcNAc should be GlcNAc without the "N".

REVIEWER COMMENTS

Reviewer #1 (Remarks to the Author):

The crystal structure of the Arabidopsis nucleocytoplasmic protein O-fucosyltransferase SPY is reported. The work provides mechanistic insights into SPY function by identifying amino acids involved in protein O-fucosylation. SPY was found to self-modify two serines located near the N-terminus. Deletion of the region containing these residues reduces SPY activity suggesting that self-modification may regulate activity. The results generally support the authors' conclusions. The work provides an important foundation for increasing our understanding of the glycosyltransferase family 41 function and how these enzymes regulate plant development.

We thank the reviewer for the concise summary and insightful comments on our work.

Comments:

Line 28 and lines 249-251. The data does not support that O-fucosylation negatively regulates SPY. Since the role of self-fucosylation was tested by deleting the region that is modified (Fig. 2d), the results only show that AA 1-42 negatively regulate SPY. To more directly address the role of O-fucose modification, the activity of the S21A, S24A mutants and an S21A / S24A double mutant should be determined.

Our response: We apologize for having misled this reviewer and thank the reviewer for the suggestion. Our data reveal that SPY Δ^{1-42} but not SPY Δ^{1-14} shows higher activity compared with the full-length SPY (Fig. 2d). Meanwhile, we showed that fucosylation modification occur in the region between V15 and Q42 of SPY (Fig. 2b, c). Therefore, we concluded ‘...SPY’s self-fucosylation region negatively regulates its enzyme activity...’. Here, self-fucosylation region refers to V15-Q42 of SPY. Following this reviewer’s suggestion, we further measured the activity of SPY^{S21A}, SPY^{S24A} and SPY^{S21A/S24A} toward DELLA¹⁻²⁰⁵ (new Fig. 2e). All the three mutants show similar activity compared with wild type SPY. Taken together, our data demonstrate that the self-fucosylation region (V15-Q42) but not O-fucosylation on SPY can negatively regulate its activity. The results have been presented in Page 12 lines 253-258.

Lines 125-127. While the hydrolysis assay supports that C645S SPY is a fully functional enzyme, the evidence would be stronger if C645S SPY was shown to have wild-type activity toward a protein substrate such as DELLA.

Our response: we thank the reviewer for the suggestion. We have included new data showing that mutation of C645 to Ser does not impair SPY's activity toward DELLA¹⁻²⁵⁰ (new Supplementary Fig. 3b). The data have been described in Page 7 lines 128-134.

Lines 125-127. Consider reinforcing the argument that C645 is a surface residue by highlighting its location on a structure shown in one of the figures. Consider pointing out that C645S is not conserved (Supp Fig 10), which further supports the argument that mutating it will not affect activity.

Our response: we thank the reviewer for the suggestions. We have shown C645S in ball-and-stick model in new Figure 1b to highlight its location. We have also pointed out that C645 in *Arabidopsis* SPY is not conserved in Page 6 lines 118-121.

Line 284. I cannot see any signal for SPYN5 and N7 in the figure. Is there a signal or is it undetectable?

Our response: We apology for the confusion. We repeated the experiment for more than three times. Even after long exposure, the signal for SPY^{N5} and SPY^{N7} were undetectable. We clarified this issue in the revised manuscript (Page 14 line 291).

A number of SPY missense mutations affecting plant hormone and other responses are reported in the literature. It would increase the impact and utility of this work if the authors showed where these mutation map on the structure and discuss if the locations suggest that enzyme activity is affected.

Our response: we thank this reviewer for the suggestion. We have provided a figure in which the SPY missense mutations are highlighted on the structure (new Supplementary Fig. 12). We have also discussed whether the SPY missense mutations would affect the activity in the revised manuscript (Pages 24-26 lines 537-576).

Minor comments:

Line 113. serial should be series

Our response: We have corrected the typo and thank the reviewer for pointing this out.

Line 118. Please spell out size exclusion chromatography multi angle light scattering (SEC-MALS).

Our response: We revised it as suggested.

Line 158. Consider large rather than dramatical.

Our response: We revised it as suggested.

Line 201 Unlike should be unlikely.

Our response: We revised it as suggested.

Line 222. Replace can with that.

Our response: We revised it as suggested.

Line 224. Delete truly.

Our response: We revised it as suggested.

Line 232. ... modification occur in the region between V15 and Q42.

Our response: We revised it as suggested.

Line 263. Delete excellently.

Our response: We revised it as suggested.

Reviewer #2 (Remarks to the Author):

This is a very nice manuscript that describes the first crystal structure of SPY, a plant fucosyltransferase that fucosylates protein substrates. The relevance of this structure

is of utmost importance because it mainly describes for the first time how the TPRs interact with potential protein substrates (in this case SPY itself). This has never been visualized at the atomic level, and this structure exemplifies this. Therefore, this work offers a significant novelty to merit publication in Nature Communications. However, major and minor comments would need to be addressed to improve the quality of the manuscript.

Our response: We thank this referee for the insightful comments and the recognition of our work.

Major comments:

- The structure raises a major question that needs to be addressed. For example, how does the dimer behave in the presence of protein substrates? This needs to be addressed to understand this enzyme's behavior in solution and to rule out whether a monomeric or other form might be present. Their trapped structure might suggest that it is likely an inactive conformation of the enzyme leading to self-fucosylation. Would the dimer be the functional form to recognize protein substrates? Biophysical experiments could show this by incubating SPY with different protein substrates that need SPY TPRs for optimal interaction. For example, they could use PRR5 as a protein substrate. In addition, if the dimer holds in the presence of PRR5, would this be 2:2 (SPY:protein substrate) stoichiometry or 2:1 stoichiometry?

Our response: Thanks for bringing this up. We have done systematical effort using bacterial (*E. coli* cell), insect (Sf9 and Hi 5 cells) and mammalian (HEK 293F cell) expression systems but unfortunately could not get recombinant PRR5 protein. Previous study¹ and our data both have shown that SPY could fucosylate DELLA¹⁻²⁰⁵ (Fig. 2d). Therefore, we checked the oligomerization state of SPY in presence of DELLA¹⁻²⁰⁵ using gel filtration (new Supplementary Fig. 6d). Wild type SPY remains a dimer in solution after incubated with DELLA¹⁻²⁰⁵. SPY^{H495A}, which is a catalytically dead mutant, also adopts dimeric form both in absence as well as in presence of DELLA¹⁻²⁰⁵ and GDP-fucose. Altogether, our data reveal that dimer is the functional form of SPY. These data have been described in Page 21 lines 450-459.

To investigate the SPY: protein substrate stoichiometry in the catalytic reaction, we prepared a half-dead heterodimer with one protomer being wild type SPY (SPY^{WT}) and the other one being a catalytical dead mutant, SPY^{K665A} (see details in methods). If it is 2:1 stoichiometry, the SPY^{WT}/SPY^{K665A} heterodimer and the SPY^{WT}/SPY^{WT} homodimer would show similar activity. In fact, compared with the SPY^{WT}/SPY^{WT} homodimer, the SPY^{WT}/SPY^{K665A} heterodimer shows substantially

weaker activity toward itself and DELLA¹⁻²⁰⁵ (new Supplementary Fig. 6e,f), demonstrating that SPY catalyzes O-fucosylation of protein substrates in 2:2 stoichiometry. These data have been described in Page 21 lines 459-463.

- The authors should check the oligomerization state of SPY once it fucosylates itself. Will the self-fucosylated SPY still be a dimer? Also, check the oligomerization state of SPY truncations and mutants S21A and S24A. Will these mutants and truncations still be a dimer?

Our response: We thank this reviewer for the suggestions. We performed gel filtration assay to show that self-fucosylated SPY, SPY^{Δ1-14}, SPY^{Δ1-42}, SPY^{S21A}, SPY^{S24A} and SPY^{S21A/S24A} are dimers in solution (new Supplementary Fig. 6c). These data have been described in Page 21 lines 450-459.

- Define SPYN5 and SPYN7 in the text, not only in the figure legend. The experiments with these mutations (the asparagine ladder mutations) are very nice and exemplify previous biophysical experiments of OGT with similar Asn residues as crucial in recognition of protein substrates. The authors should check if these mutations also kill the activity on other protein substrates such as PRR5 or large protein substrates. Check also the activity of K231 or D266 mutants on protein substrates such as PRR5.

Our response: We apology for the confusion and thank this reviewer for the constructive suggestions. We defined SPY^{N5} and SPY^{N7} in Page 14 lines 287-288. We agree with the reviewer that it will be interesting to investigate if the SPY^{N5} and SPY^{N7} lost activity towards PRR5. However, as mentioned above, we could not get recombinant PRR5 protein and thus were unable to perform the *in vitro* enzymatic assay with PRR5. Our data showed that the activity of SPY^{N5} and SPY^{N7} toward itself and DELLA¹⁻²⁰⁵ were completely abrogated (new Fig. 3b,c). Considering the full-length SPY and DELLA¹⁻²⁰⁵ are both large protein substrates and they share low sequence similarity, we hope this reviewer would agree that our data have largely demonstrated the importance of the asparagine ladder.

Following the reviewer's suggestion, we evaluated the activity of SPY^{K231A}, SPY^{D266A}, SPY^{K231A/D266A}. Unexpectedly, mutations of K231 and D266 to alanine did not affect the self-fucosylation but had a significant impact on the fucosylation of DELLA¹⁻²⁰⁵ (new Fig. 3b,c). SPY^{K231A} and SPY^{K231A/D266A} showed substantially decreased activity while SPY^{D266A} showed increased activity toward DELLA¹⁻²⁰⁵.

Meanwhile, compared with SPY^{K231A}, the double mutant showed increased activity. Taken together, our data demonstrated that K231 and D266 drive SPY's protein substrate selection. These results have been described in Page 14 lines 301-309 and discussed in Page 23 lines 496-503.

- Are these Asn residues conserved with the Asn residues in the human OGT (see PMID: 33709700)? I guess not but it would be nice to compare the positions of these Asn residues in SPY with the Asn residues in the human OGT.

Our response: We thank this reviewer for the constructive suggestion. In the revised manuscript, we performed sequence alignment of the TPRs in *Arabidopsis* SPY and human OGT (new Supplementary Fig. 5d). Most of these Asn residues are conserved and occupy position 6 in the TPR consensus. Yet at the same time, different from the asparagine ladder in human OGT which stretches across almost the entire TPR region (TPR 3-TPR 14), that in *Arabidopsis* SPY threads through a relatively smaller region (TPR 5-TPR 11). Besides, N333 and N367 in *Arabidopsis* SPY occupying position 5 in TPR 9 and TPR 10 also contribute to the recognition of substrate loop, whereas the equivalent residues in human OGT (S355 and S389) are not conserved and do not interact with the substrate peptide. These data have been described in Page 22 lines 477-486.

- The mechanism for SPY is not clear. The authors suggest that H495 is the catalytic base. Is this residue conserved with the human OGT? Note that for the human OGT, it was proposed earlier that a His residue was the catalytic base. Then, two further manuscripts suggested that the alpha phosphate (PMID: 23103942) or a chain of water molecules in which one of them interacts with an Asp residue (PMID: 23103939) could act as potential catalytic bases. This should be discussed in the manuscript and the H495 should be clarified if it occupies the same position or similar positions to His498 or His558 in the human OGT. Therefore, new figures showing a proper comparison with the human OGT active site should be shown in the manuscript. In addition, they should compare all these mechanisms with the ones described for PoFUT1, PoFUT2, FUT8, etc (PMIDs: 26854667, 34868727 and 32080177). SPY as a PoFUT should be compared with very distant PoFUTs such as PoFUT1 and PoFUT2 and other fucosyltransferases (FUT8), which also interact with proteins. Finally, molecular dynamics simulations should be performed with GDP-fucose and SPY-peptide to determine the distance of H495 to the acceptor Ser residue of the peptide during the simulations.

Our response: We thank this reviewer for the constructive suggestions. H498, H558 and Y841 in human OGT were initially proposed as candidate bases²⁻⁵, but were later observed to locate too far away from the acceptor hydroxyl^{6,7}. D554 in human OGT was put on the stage because it indirectly interacted with the acceptor hydroxyl via a chain of water molecules⁷, however, human OGT^{D544A} retained activity⁶. Following this reviewer's suggestion, we performed structural superimposition of *Arabidopsis* SPY and human OGT using the catalytic domains as references and provided a new figure showing the active sites in the two proteins in parallel (new Supplementary Fig. 5e). The equivalent residues of H498, D554, H558 and Y841 in human OGT are L439, D491, H495 and A664 in *Arabidopsis* SPY, respectively. Apparently, leucine and alanine could not enable the deprotonation of the acceptor hydroxyl. D491 and H495 are adjacent in the structure while the imidazole side chain of H495 stacks with the D491 carboxylate, indicating that H495 and D491 may function as a catalytic dyad. Previously we showed that mutation of H495 to alanine abrogate the enzyme activity. In the revision, we further test two mutants, SPY^{D491A} and SPY^{H495F}. Although mutation of H495 renders the enzyme inactive, mutation of D491 does not abrogate the activity. Considering mutation of H495 to alanine would free up more space in the active site while phenylalanine is more hydrophobic and slightly bigger than histidine, it is possible that the effects of the H495 mutations are due to structural reasons. As suggested by this reviewer, we have reprocessed the diffraction data to a resolution of 2.85Å with the CC1/2 in the highest resolution being 0.527. During structure refinement with the new data, we noted that the density map for S21* hydroxyl is poor (Response Fig. 1). It would be more reasonable that the S21* hydroxyl takes an alternative rotamer (rotamer 2) to be properly aligned for attack on the anomeric carbon (Response Fig. 1 and new Supplementary Fig. 5e). On this occasion, the S21* hydroxyl rotamer is not suitable for engaging in a hydrogen bond with either the D491 carboxylate or the H495 imidazole ring. All these data demonstrate that H495 and D491 would be unfit for the role as general base. Further inspection of the 'catalytic SPY'/GDP/'substrate SPY' complex reveals that none of the residues in SPY is within 4 Å of S21* hydroxyl. Altogether, SPY may not harbor the catalytic base. Notably, the acceptor substrate hydroxyl donates a hydrogen bond to one of the α -phosphate oxygen in the donor substrate while the α -phosphate lacks any interactions with positively charged side chains, hence this phosphate could serve as the catalytic base (new Fig. 4e). In fact, a similar substrate-assisted catalysis mechanism has been recently revealed for human OGT based on cumulative crystallographic snapshots and biochemical probes⁶, which coincidentally reinforces

SPY's substrate-assisted catalysis proposed here. We understand that more studies on trapping a complex of SPY and intact substrates by reducing the rate of enzymatic turnover in crystallo using artificial substrate analogs probably could eventually elucidate the catalytic mechanism. We hope this reviewer would agree that such efforts could be a future following-up project.

In the revised manuscript, we described the new mutagenesis data, the new mechanistic scenario for SPY (Pages 16-17 lines 348-367), and compared the catalytic mechanism for SPY with the ones for human OGT, PoFUT1, PoFUT2 and FUT8 (Pages 23-24 lines 508-536).

Response Fig. 1 Stereo view of the active site in *Arabidopsis* SPY. The *Fo-Fc* electron density (2.0σ level) is shown for the substrate loop (in green). Three possible rotamers of S21* is shown in the figure. Previously the S21* hydroxyl was placed in rotamer 1, which is not aligned for attack on the anomeric carbon. Apparently, rotamer 2 is properly aligned for attack. Meanwhile, the α -phosphate oxygen is hydrogen bonded to the hydroxyl group of rotamer 2.

- According to the crystallography table, the structure is of lower resolution since the $CC1/2$ at high resolution is around 0.2. The cutoff for the highest resolution shell should have a $CC1/2$ around 0.5. Therefore, the data must be rescaled to render a $CC1/2$ around 0.5 in the highest resolution shell. This will clearly decrease the current resolution, but it will reflect better the resolution of the structure.

Our response: We have reprocessed the diffraction data to a resolution of 2.85Å. The $CC1/2$ in the highest resolution is 0.527 (new Table 1). Accordingly, we have re-refined the structure using the new data. Particularly, we noted that the density map for S21* hydroxyl is poor and thus corrected the rotamer of S21* hydroxyl to be

properly aligned for attack on the anomeric carbon (new Supplementary Fig. 9 and Response Fig. 1).

- Show the maps for the Fo-Fc in Supplementary Figure 3. This map is more relevant to see the quality of the density.

Our response: We have shown the *Fo-Fc* map for GDP in new Supplementary Fig. 4. Additionally, we have also shown the *Fo-Fc* map for the N-terminal loop in new Supplementary Fig. 9.

- Discussion is too short and should be enlarged, taking into account the catalytic mechanism and also exploiting that this structure is the first one showing interactions between the TPRs and a peptide (here, the N-terminus of SPY).

Our response: We have enlarged the Discussion as suggested.

Minor comments:

- Self-glycosylation is not novel since this occurs in many glycosyltransferases such as human OGT, NleB1, etc. See, e.g., PMID: 32411621. Yet, their finding is very interesting. Cite some of these papers.

Our response: Thanks for pointing this out. We have cited three related papers^(5,8,9) and discussed along with our results (Page 20, lines 432-436).

- For the ordered bi-bi catalytic mechanism, the authors need to perform additional kinetic experiments and/or NMR experiments to demonstrate this mechanism.

Our response: We completely agree with the reviewer that it will be interesting to experimentally demonstrate the ordered bi-bi catalytic mechanism. In this case, kinetic experiments need to be performed in the presence of GDP at saturating protein substrate concentration while varying GDP-fucose levels^{10,11}. However, the protein substrates for SPY including the DELLA¹⁻²⁰⁵ and catalytically dead mutants of SPY (SPY^{H495A} and SPY^{K665A}) precipitate easily at a concentration above 0.2 mM. Besides, we were unable to develop an assay for sensitively and quantitatively measuring SPY's glycosyltransferase activity. Considering SPY could also hydrolyze GDP-fucose, monitoring the amount of GDP in the reaction system is not a suitable way to measuring SPY's glycosyltransferase activity. Meanwhile, neither the

radiolabeled GDP-fucose nor the high-quality antibodies that specifically recognize O-linked fucose are commercially available. Hampered by these factors, at present we are unable to proceed the kinetic experiments as suggested. On the other hand, SPY exists as dimers in solution with a molecular weight of about 200 kD, which is too big for the NMR analysis. We are sorry to say that this is the limitation of our work. In fact, a similar structural feature was observed in the ternary complex of the closest structural homolog, human OGT (Fig. 4b and Supplementary Fig. 5e). Reinforced by kinetic studies with radiochemical UDP-¹⁴C-GlcNAc as the glycan donor, the ordered bi-bi catalytic mechanism has been demonstrated for human OGT². In the revised manuscript, we described the comparison of SPY and human OGT (Page 16 lines 341-347) and erased the ordered bi-bi catalytic mechanism from the schematic diagram of catalysis (new Fig. 4e).

- sentence in lines 37 and 38 does not make much sense. “Different from secretion proteins”. Rewrite this sentence.

Our response: We apology for the confusion. We have rewritten this sentence in the revision (Page 3, lines 34-38).

- What do they mean with aberrant O-linked monosaccharide glycosylation and its linking to diabetes and other diseases? Aberrant or truncated O-glycans are linked to O-glycosylation in the Golgi or E.R. pathway and not to O-glycosylation in the cytosol or in the nucleus. Clarify this.

Our response: Thanks for pointing this out. We have corrected it in the revision (Page 3, lines 38-41).

- Line 65. Define DELLAs. Define also GA signal transduction (line 55).

Our response: We revised it as suggested.

- Supplementary Figure 11. N-GlcNAc should be GlcNAc without the “N”.

Our response: We have corrected the typo and thank the reviewer for pointing this out.

References:

- 1 Zentella, R. *et al.* The Arabidopsis O-fucosyltransferase SPINDLY activates nuclear growth repressor DELLA. *Nat Chem Biol* **13**, 479-485, doi:10.1038/nchembio.2320 (2017).
- 2 Lazarus, M. B., Nam, Y., Jiang, J., Sliz, P. & Walker, S. Structure of human O-GlcNAc transferase and its complex with a peptide substrate. *Nature* **469**, 564-567, doi:10.1038/nature09638 (2011).
- 3 Martinez-Fleites, C. *et al.* Structure of an O-GlcNAc transferase homolog provides insight into intracellular glycosylation. *Nat Struct Mol Biol* **15**, 764-765, doi:10.1038/nsmb.1443 (2008).
- 4 Dorfmüller, H. C. *et al.* Substrate and product analogues as human O-GlcNAc transferase inhibitors. *Amino acids* **40**, 781-792, doi:10.1007/s00726-010-0688-y (2011).
- 5 Clarke, A. J. *et al.* Structural insights into mechanism and specificity of O-GlcNAc transferase. *EMBO J* **27**, 2780-2788, doi:<https://doi.org/10.1038/emboj.2008.186> (2008).
- 6 Schimpl, M. *et al.* O-GlcNAc transferase invokes nucleotide sugar pyrophosphate participation in catalysis. *Nat Chem Biol* **8**, 969-974, doi:10.1038/nchembio.1108 (2012).
- 7 Lazarus, M. B. *et al.* Structural snapshots of the reaction coordinate for O-GlcNAc transferase. *Nat Chem Biol* **8**, 966-968, doi:10.1038/nchembio.1109 (2012).
- 8 Pan, X., Luo, J. & Li, S. Bacteria-Catalyzed Arginine Glycosylation in Pathogens and Host. *Front Cell Infect Microbiol* **10**, 185, doi:10.3389/fcimb.2020.00185 (2020).
- 9 Capotosti, F. *et al.* O-GlcNAc transferase catalyzes site-specific proteolysis of HCF-1. *Cell* **144**, 376-388, doi:10.1016/j.cell.2010.12.030 (2011).
- 10 Copeland, R. A. *Enzymes : a practical introduction to structure, mechanism, and data analysis*. Vol. xvi 397p (Wiley, New York, 2000).
- 11 Segel, I. H. *Enzyme Kinetics. Behavior and Analysis of Rapid Equilibrium and Steady-State Enzyme Systems*. Vol. xxii 957p (Wiley, New York, 1975).

REVIEWERS' COMMENTS

Reviewer #1 (Remarks to the Author):

The concerns with the previous submission are addressed in this version but there are a few comments/suggestions.

Comments:

Line 53: GA is the abbreviation for gibberellin not gibberellic acid, which is GA3.

Line 130: Taking rather than Taken

Line 134: delete respectively

Lines 156-157: This sentence doesn't convey the argument that C645 is surface exposed and located on a loop connecting the N-Cat and C-Cat providing further evidence that the crystal structure of engineered SPYC645S represents the wild type SPY conformation. Also the statement in this location disrupts the narrative. You could consider moving this argument up to the other discussion of this mutation or just delete it since the arguments you make above make a strong case.

Line 291: delete either

Line 460-463: Can you rule out that the extra steps used to purify the SPYWT/SPYK665A heterodimer did not reduce its specific activity? This seems possible. To control for this you could use the same strategy to prepare SPYWT homodimer. Also, was the experiment repeated with multiple enzyme preparations?

Line 565: Consistent rather than In consistent.

Reviewer #2 (Remarks to the Author):

The authors have done a great job and have responded to all my questions. Therefore, in my opinion, the manuscript in its current form is suitable for NCOMMS.

Nevertheless, I have some minor changes:

- Page 21. Replace demonstrating by suggesting. To demonstrate the stoichiometry, authors should perform ITC experiments.

- In Fig. 5d, for the alignment of TPRs, I cannot see the Asn residues forming the ladder in SPY. E.g., Asn300 in the manuscript appears to be Asn302 in the alignment. Double-check this because I have problems identifying the Asn residues mentioned in the manuscript in the alignment.

- The mechanism in Figure 4e does not look right. SPY is an inverting FT, and SPY appears to be a retaining FT in the mechanism depicted by the authors. Check also the fucose moiety because it is wrong.

- Line 349-350: mention Asp, Glu and His as potential catalytic bases in the reaction mechanism. Replace His/Glu by just only His.

- Line 367: finish the sentence as indicated below,

“the catalytic base as proposed earlier for human OGT (reference XXX)”.

- Line 510: add “likely” in-between “but” and “by the...”.

- Line 457 and 458. It is not clear to me whether the authors can say that the dimer stays as a dimer in the presence of the protein substrate because by gel filtration the enzyme and the protein substrate do not coelute together. This is likely due that the protein substrate has a poor affinity for SPY. Reconsider writing this paragraph to tone down the claims.

REVIEWER COMMENTS

Reviewer #1 (Remarks to the Author):

The concerns with the previous submission are addressed in this version but there are a few comments/suggestions.

We thank the reviewer for the positive comment.

Comments:

Line 53: GA is the abbreviation for gibberellin not gibberellic acid, which is GA3.

Our response: Thanks for pointing this out. We have corrected it in the revision.

Line 130: Taking rather than Taken

Our response: We revised it as suggested.

Line 134: delete respectively

Our response: We revised it as suggested.

Lines 156-157: This sentence doesn't convey the argument that C645 is surface exposed and located on a loop connecting the N-Cat and C-Cat providing further evidence that the crystal structure of engineered SPYC645S represents the wild type SPY conformation. Also the statement in this location disrupts the narrative. You could consider moving this argument up to the other discussion of this mutation or just delete it since the arguments you make above make a strong case.

Our response: Thanks for pointing this out. We have deleted this sentence in the revision.

Line 291: delete either

Our response: We revised it as suggested.

Line 460-463: Can you rule out that the extra steps used to purify the SPYWT/SPYK665A heterodimer did not reduce its specific activity? This seems possible. To control for this, you could use the same strategy to prepare SPYWT homodimer. Also, was the experiment repeated with multiple enzyme preparations?

Our response: We apology for the confusion. In this experiment, we did prepare SPY^{WT}/SPY^{WT} homodimer using the same strategy as that for the SPY^{WT}/SPY^{K665A} heterodimer. We have clarified this issue in the revision (Page 28 lines 609-614).

We have repeated this experiment with more that 3 enzyme preparations and always got similar results.

Line 565: Consistent rather than In consistent.

Our response: We revised it as suggested.

Reviewer #2 (Remarks to the Author):

The authors have done a great job and have responded to all my questions. Therefore, in my opinion, the manuscript in its current form is suitable for NCOMMS.

We thank the reviewer for the positive comment.

Nevertheless, I have some minor changes:

- Page 21. Replace demonstrating by suggesting. To demonstrate the stoichiometry, authors should perform ITC experiments.

Our response: We revised it as suggested.

- In Fig. 5d, for the alignment of TPRs, I cannot see the Asn residues forming the ladder in SPY. E.g., Asn300 in the manuscript appears to be Asn302 in the alignment. Double-check this because I have problems identifying the Asn residues mentioned in the manuscript in the alignment.

Our response: Thanks for pointing this out. We have corrected the numbering in new Supplementary Fig. 5d.

- The mechanism in Figure 4e does not look right. SPY is an inverting FT, and SPY appears to be a retaining FT in the mechanism depicted by the authors. Check also the fucose moiety because it is wrong.

Our response: Thanks for pointing this out. We have corrected them in the revision (new Fig. 4e)

- Line 349-350: mention Asp, Glu and His as potential catalytic bases in the reaction mechanism. Replace His/Glu by just only His.

Our response: We revised it as suggested.

- Line 367: finish the sentence as indicated below, "the catalytic base as proposed earlier for human OGT (reference XXX)".

Our response: We revised it as suggested.

- Line 510: add “likely” in-between” but” and “by the...”.

Our response: We revised it as suggested.

- Line 457 and 458. It is not clear to me whether the authors can say that the dimer stays as a dimer in the presence of the protein substrate because by gel filtration the enzyme and the protein substrate do not coelute together. This is likely due that the protein substrate has a poor affinity for SPY. Reconsider writing this paragraph to tone down the claims.

Our response: Thanks for the comments. We revised it as suggested (Page 21 lines 456-462).